# Fully Automatic Segmentation of 3D Brain Ultrasound: Learning from Coarse Annotations

**Rüdiger Göbl** *
Computer Aided Medical Procedures
Technical University Munich

**Julia Rackerseder** *
Computer Aided Medical Procedures
Technical University Munich

**Nassir Navab**
Computer Aided Medical Procedures
Technical University Munich
Computer Aided Medical Procedures
Johns Hopkins University

**Christoph Hennersperger**
Computer Aided Medical Procedures
Technical University Munich

## Abstract

Intra-operative ultrasound is an increasingly important imaging modality in neurosurgery. However, manual interaction with imaging data during the procedures, for example to select landmarks or perform segmentation, is difficult and can be time consuming. Yet, as registration to other imaging modalities is required in most cases, some annotation is necessary. We propose a segmentation method based on DeepVNet and specifically evaluate the integration of pre-training with simulated ultrasound sweeps to improve automatic segmentation and enable a fully automatic initialization of registration. In this view, we show that despite training on coarse and incomplete semi-automatic annotations, our approach is able to capture the desired superficial structures such as *sulci*, the *cerebellar tentorium*, and the *falx cerebri*. We perform a five-fold cross-validation on the publicly available RESECT dataset. Trained on the dataset alone, we report a Dice and Jaccard coefficient of $0.45 \pm 0.09$ and $0.30 \pm 0.07$ respectively, as well as an average distance of $0.78 \pm 0.36 \ mm$. With the suggested pre-training, we computed a Dice and Jaccard coefficient of $0.47 \pm 0.10$ and $0.31 \pm 0.08$, and an average distance of $0.71 \pm 0.38 \ mm$. The qualitative evaluation suggest that with pre-training the network can learn to generalize better and provide refined and more complete segmentations in comparison to incomplete annotations provided as input.

## 1 Introduction

Intra-operative ultrasound (iUS) is becoming increasingly important in neurosurgery. This can be related to its ability of real-time imaging and the lack of ionizing radiation. In open brain surgery, ultrasound can help the surgeon navigate, as well as relate pre-operative images, like magnetic resonance imaging (MRI), to facilitate additional information, such as structural detail and resection treatment plans.

To be able to fully exploit the advantages of iUS, a fusion of this intra-operative information with pre-interventional (tomographic) imaging data is important to optimally support clinical staff during the surgery. While there are several works addressing the task of image-based registration of iUS with pre-operative imaging, they require adequate initialization, due to their prohibitive difficulties to

---

*Both authors contributed equally to this work

1st Conference on Medical Imaging with Deep Learning (MIDL 2018), Amsterdam, The Netherlands.

converge in the presence of large misalignment. In the neurosurgical context, the capture range is reported to be as low as $15\ mm$ [7], leading to the necessity of a sufficiently high initial overlap.

Although tracking data is available for initialization during surgery itself, such information usually is not stored for retrospective studies. In this case, standard of practice is to use landmark based rigid registration as a first step. However, even considering the ever increasing quality of ultrasound data, interpretation of iUS can be intricate, and surgeons are often not trained experts in this modality. Therefore, any manual interaction like landmark selection or feature extraction is prone to error, which is reflected in rather high reported inter- and intra-observer variability of up to 1.6 mm [14]. Instead, it can be advantageous to segment correlating structures in both modalities that can then be used for initialization [2] or even registration [17].

Another challenge in neurosurgery is the inevitable movement of the brain after opening skull and dura mater, called brain shift. The displacement caused when opening the *dura mater* alone is reported to be up to $13.4\ mm$, increasing constantly over time [3]. Although iUS is used to manage brain shift [19], Gerard *et al.* [8] report that surgeons trust image fusion with pre-operative data less and less over the course of the intervention. In other words, this means that brain shift can render registration through tracking data useless.

A possible solution to this may also be the segmentation of anatomical structures in order to track the movement of the brain and allow for re-registration during the surgery. In both outlined cases, the segmented structures need to be clearly visible from most angles and even after brain shift has occurred. As described in a previous work by our group [2], it can be useful to use *lateral ventricles*, but also rather superficial structures such as *sulci*, mainly the prominent *central* and *precentral sulcus* and of course as already shown by Nitsch *et al.* [17] the *cerebellar tentorium*, and the *falx cerebri* which we expand towards the *transverse* and *longitudinal fissure*, respectively. The latter is especially useful when battling brain shift, since only minor shift in the midline is reported [3, 8].

## 1.1 State-of-the-art

Automatic segmentation of different brain structures in ultrasound has been addressed by several works over the last years to circumvent the subjectiveness and extensive use of time when labeling manually [15]. These problems often lead to insufficient, incomplete or coarse delineation, like in the semi-automatic segmentation used for annotations in our work. Velásquez-Rodríguez *et al.* [22] segment the whole cerebellum in fetal ultrasound using a spherical harmonics model and gray level voxel profiles to support experts in this challenging task. Another group [18] segments ventricles in neonatal 3D US with the help of a multi-atlas initialized multi-region segmentation approach.

In adult transcranial 3D ultrasound (3D TCUS) Ahmadi *et al.* [1] segmented the midbrain based on a statistical shape model with only minimal user interaction and later Milletari *et al.* [15] expanded that work using sparse autoencoder and Hough Forest to be fully automatic and applicable in multiple anatomies, such as left ventricle and prostate. Recently, it was shown that Hough voting can also be used in a deep learning setting to solve the same task of midbrain segmentation in 3D TCUS [16] with similar to better Dice score, depending on the amount of training data used.

In 3D intra-operative brain ultrasound, a patient-specific model is used to segment tumors. Ilunga-Mbuyamba *et al.* [11] use hyperechogenic structures extracted from MRI and US with the Otsu method to guide the registration performed with normalized gradient field. However, usually only cystic tumors manifest as hyperechogenic in ultrasound, therefore this method is not generalizable. As already described above, Nitsch *et al.* [17] segment falx cerebri and its neighboring gyri in 2D ultrasound with the goal to facilitate registration with pre-operative MRI. This method is also not generally applicable, since the segmented structure is only visible in certain surgical settings.

Although undeniably deep learning approaches have taken the world of medical imaging by storm, only few groups have utilized deep learning in ultrasound segmention so far. A recent review on machine learning advances in ultrasound [4] lists only few examples of segmentation in this modality. By means of deep learning Brattain *et al.* list only segmentation of the left ventricle, lymph node, placenta and midbrain. The latter ist discussed in more detail above. Salehi *et al.* [21] use a fully convolutional neural network (FCNN) to segment bone surface in ultrasound, which is then used for image registration with pre-operative computed tomography. Most other methods focus around obstetrics, most probably because in this field ultrasound is used almost exclusively in terms of imaging and 3D US is already clinical standard in many countries. Looney *et al.* [13] segmented the

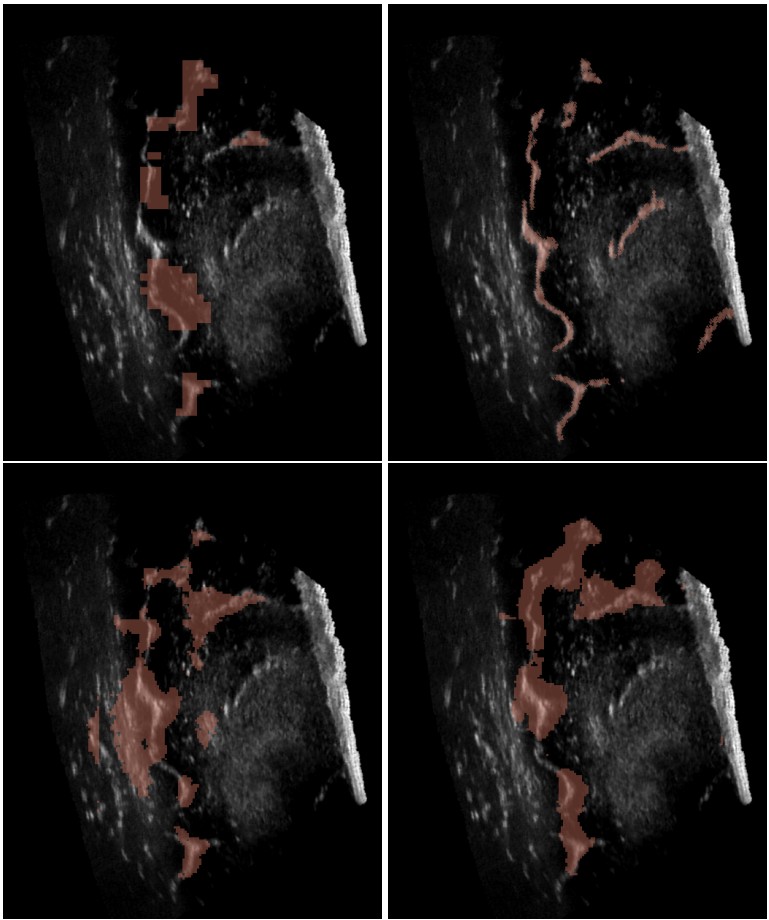

Figure 1: **Segmentation example** Patient number 27, segmentation in red superimposed on ultrasound image in original resolution. From left to right, top to bottom: coarse semi-automatic segmentation used for training, fine manual segmentation, predicted segmentation DenseVNet (*Scratch*), predicted segmentation pre-trained DenseVNet (*Fine-tuned*)

placenta from 3D ultrasound using DeepMedic [12], a multi-scale 3D Deep Convolutional Neural Network coupled with a 3D fully connected Conditional Random Field for brain lesion segmentation. Another rather broad approach detects anatomical structures and segments them in US images. The method is shown on the example of left ventricle segmentation and head circumference in obstetric exams [5].

## 2 Methods

In order to facilitate fully automatic US-MRI registration initialization, a problem discussed in Sec. 1.1, we propose a segmentation method for 3D intracranial US, based on the DenseVNet architecture [9] that was developed for the segmentation of abdominal CT. We combine this with a pre-training simulation based method in order to ease limitations in dataset size, which is due to the database used (see Sec. 2.1) and common for learning in clinical settings.

### 2.1 Dataset preparation

To ensure reproducibility of our work, we use the publicly available RESECT dataset [23], a retrospective study on 23 low-grade glioma patients who underwent open brain surgery. It is comprised of co-registered FLAIR and T1 MRI sequences, as well as three reconstructed 3D-US

volumes from before, during and after resection for each patient. In this paper we use T1 MRI and pre-resection ultrasound. We keep the numbering of subjects according to the original publication.

**MRI**

In order to be able to simulate ultrasound sweeps from MRI (discussed in Sec. 2.2) we employ FreeSurfer[2] [6] for skull stripping and subcortical segmentation. The resulting tissue maps contain information about white and gray matter, cerebrospinal fluid as well as lateral ventricle. These are the main influencing factors for ultrasound in brain.

**Ultrasound**

A challenge in this dataset is the use of different ultrasound probes, imaging depths, and therefore diverse resulting sizes and resolution of the reconstructed volumes. We use reconstructed 3D volumes in order to avoid the accumulation of error from compounding when reconstructing the segmentation from slices to 3D (see for example Fig. 4). The reconstruction algorithm used was the built-in Sonowand Invite system's [23]. To create more homogeneous volumes for learning, we resampled them to $0.3\ mm$ isotropic voxel-spacing.

To create a comparative basis, the ultrasound volumes were processed two-fold. For training purposes, we segment *ventricles*, *sulci*, *cerebellar tentorium* and *longitudinal fissure* semi-automatically with minimal user input on down-sampled ($1\ mm$ resolution) ultrasound volumes, as described in [2]. This yields a coarse, low-resolution segmentation, which is expected from real life scenarios due to time pressure, and we show to be sufficient for training the network (Fig. 1, top left). For qualitative comparison purposes, a manual segmentation is conducted on US volumes in the original resolution to create a more refined label map (Fig. 1, top right). Manual labeling is time-consuming (up to several hours per patient), prone to inter- and intra-observer variability and puts a high mental load on the observer [15], emphasizing the challenges of this task in clinical or research practice described in Sec. 1. In concordance with this, we only have manual fine segmentations available for some datasets for visual evaluation purposes only.

## 2.2   US-simulations for pre-training

As only limited data is available, we create a simulated 3D US dataset for pre-training the model. The simulations are created with a real-time capable, hybrid ray-tracing and convolutional method [20] and are based on tissue maps of the target anatomy extracted from the MRIs as described in Sec. 2.1. The simulation allows the approximation of different imaging settings by modifying the imaging parameters and acoustic properties of the simulated tissues. Additionally, as the simulations are directly based on tissue maps, they do not require manual annotation to be used for training.

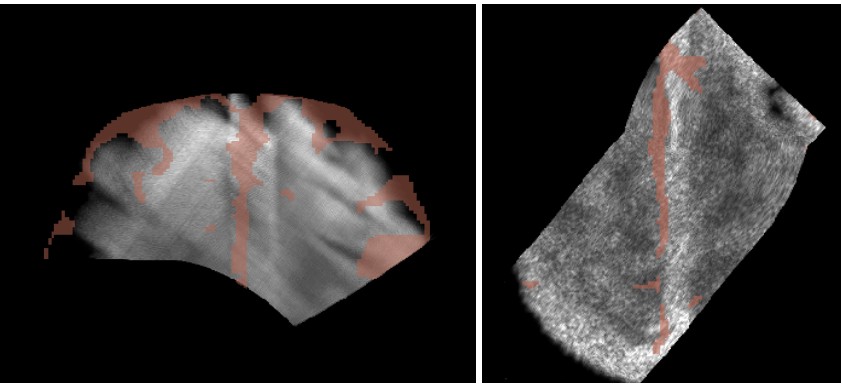

Figure 2: **Example simulations with different parametrization** based on patient number 26, underlying MRI-segmentation in red superimposed on ultrasound image.

---

[2]http://surfer.nmr.mgh.harvard.edu/fswiki/

Figure 2 shows two simulated volumes with different parametrization and their respective label-maps. The different appearance of the tissues in the simulations is clearly visible. In total, we created a dataset of 346 volumes based on patients 21, 23, 24, and 26.

## 2.3 Training

We trained the DenseVNet in the framework NiftyNet [10] with ratios of 60% for training and 20% for validation and testing sets respectively. As loss we employed the Dice-loss defined in [15], the inputs and outputs were image patches of size $128^3$.

With the memory efficient design of the DenseVNet, this allowed for a batch-size of 8. The networks were trained with the Adam optimizer for $10^5$ iterations, with a learning rate of $2 \times 10^{-5}$, $\beta_1 = 0.9$, and $\beta_2 = 0.999$. During training, we augmented the real data with random similarity transformations ($\pm 10\%$ scaling, $\pm 10°$ rotation). For the pre-trained network, $10^5$ iterations were performed on simulated volumes followed by the same number on real data.

To compare the effect of pre-training, we trained two networks: Without (*Scratch*) and with (*Fine-tuned*) pre-training. As the RESECT dataset contains only 23 volumes, we performed five-fold cross-validations for both networks.

## 3 Results

Using the coarse segmentation as input for training we inspect the performance of the resulting networks. Fig. 3 shows patient-wise results for Dice Coefficient Score (DCS) and average distance in top and bottom plot, respectively. Each subject is contained in only one test set of the five-fold cross-validation. In most cases, the measures improved when we fine-tuned the network (see Sec. 2.3), which is visualized in red compared to blue bars.

Although resulting values seem to be in the same range for most subjects, patients 1 and 21 are outliers, which is especially evident when looking at the average distance plot. Looking at Fig. 4, the explanation of this outcome becomes apparent. The left panel shows the coarse segmentation of patient 21 that we use for comparison when calculating DCS and distance. Parts of the structures that should be segmented are missing, while other parts are labeled erroneously, in comparison to the manual segmentation (middle panel). It is thus important to note that despite the fact that the Dice score is low and average distance is high for this patient, our predicted segmentation is very close to the manual labels. Although not shown, the situation is similar for patient 1.

Based on the patient-wise results, we also computed mean measurements for the *Scratch* and *Fine-tuned* networks. Comparing the predicted segmentation with the coarse labels created in a semi-automatic fashion as described in [2], which where used to train the network, results in a Dice

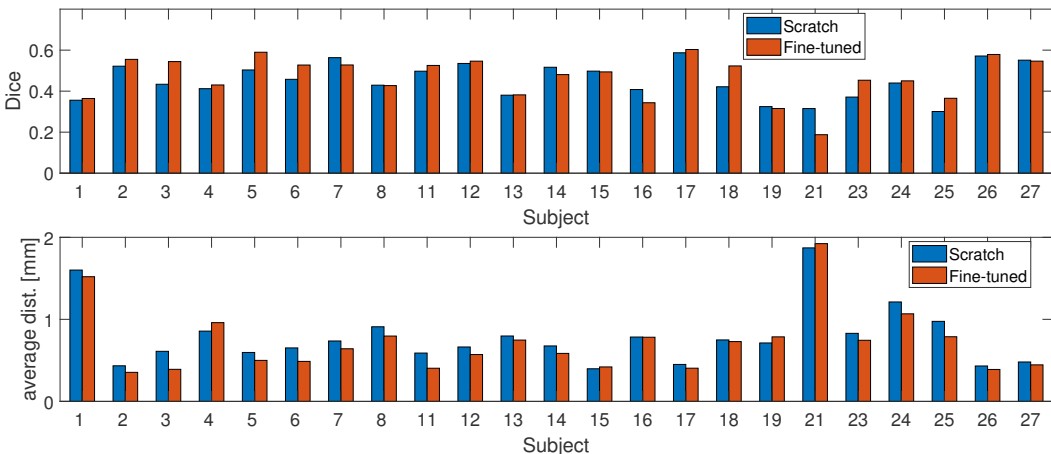

Figure 3: **Patient-wise results** for Dice Coefficient Score (top) and average distance in $mm$ (bottom) for the *Scratch* DenseVNet in blue and the *Fine-tuned* DenseVNet in red.

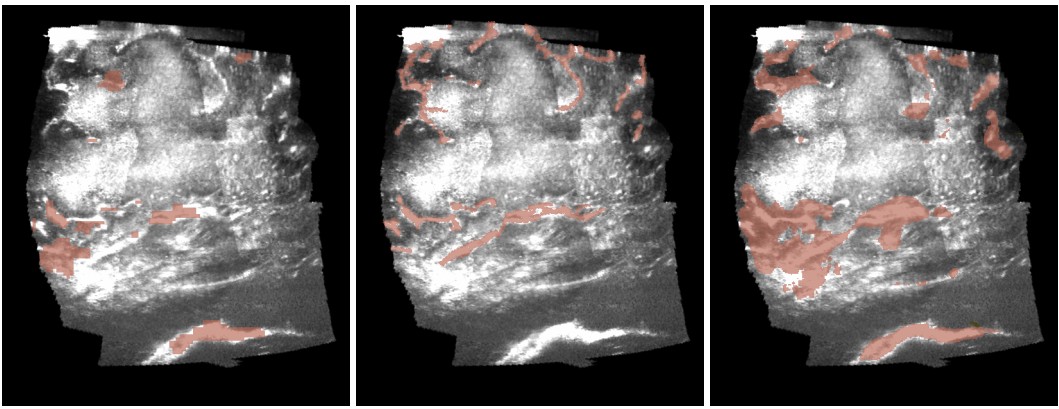

Figure 4: **Example of poor input segmentation** Segmentation for images of patient 21 in red superimposed on ultrasound image in original resolution. From left to right: coarse semi-automatic segmentation used as input for training, fine manual segmentation, predicted segmentation from fined-tuned DenseVNet.

and Jaccard coefficient of $0.45 \pm 0.09$ and $0.30 \pm 0.07$, as well as average and Hausdorff distance of $(0.78 \pm 0.36)$ $mm$ and $(21.74 \pm 5.53)$ $mm$, respectively for the DenseVNet. For the refined DenseVNet, we computed a Dice and Jaccard coefficient of $0.47 \pm 0.10$ and $0.31 \pm 0.08$, and average and Hausdorff distance of $(0.71 \pm 0.38)$ $mm$ and $(20.70 \pm 5.64)$ $mm$.

## 4   Discussion and Conclusion

Although the DCS presented here are lower than what is achieved in other segmentation tasks, the labeling predicted by the DenseVNet is sufficiently accurate for the task at hand. Examples of input for registration initialization as shown in Fig. 1 (top left) and Fig. 4 (left) are often times patchy and of poor quality due to the method they were produced with. It can be seen that in comparison to the fine (ground truth) segmentation which could be performed on few cases only, the coarse segmentation used as input for training partially labels erroneous structures for sulci and gyri. The resulting output of the proposed network, however, correctly segments the target structures and mainly overestimates the foreground labels. In this view, the results produced by the network thus cover more of the structure, that is needed for an accurate alignment with the pre-operative imaging data, than the label maps successfully used up until now.

For 3D US segmentation tasks such as the brain US application presented here, the limits in size, or availability of datasets are unfortunately not uncommon. Additionally, manual annotation thereof is difficult and time consuming, due to the high resolution of US volumes, and can take several hours per subject. This limits the availability of gold standard annotated datasets even further. To handle those limitations, we thus propose a combination of a fully convolutional deep segmentation architecture in combination with pre-training on simulations, trained on only coarse and partially incomplete semi-automatic annotations. Based on the results presented in the previous section, we are optimistic that more advanced methods can be developed to further exploit partially missing and coarse input segmentation, facilitating a generalizable segmentation on 3D-ultrasound data.

While the reported Dice coefficient scores are comparatively low, the segmentation shows more detail than the input segmentation and usually lacks less parts of the structure than the input, despite the limited dataset size of 23 patients and the coarse segmentations. This shows that DenseVNet with our training scheme is able to learn from coarse input and have more refined and complete output. For the use as input to registration algorithms, this completeness is more important than highly accurate localization and as such the results are satisfactory for interventional image processing.

In conclusion, we presented a method for fully automatic 3D brain US segmentation, that can be used for initialization of image-based registration methods. Based on the fully convolutional DenseVNet architecture in combination with pre-training on US simulations, we showed that even coarse and incomplete ground-truth segmentations can be used for training.

**Acknowledgments**

This project has received funding from the European Union's Horizon 2020 research and innovation program EDEN2020 under Grant Agreement No. 688279, as well as the GPU grant program from NVIDIA Corporation. We would like to thank ImFusion GmbH, Munich, Germany.

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
