# OpenReview forum: "Fully Automatic Segmentation of 3D Brain Ultrasound: Learning from Coarse Annotations"
_MIDL.amsterdam/2018/Conference — Submitted to MIDL 2018_

### Review · AnonReviewer2 · 2018-05-08
**The authors apply methods based on DeepVNet for automatic segmentation with (on a simulated dataset) and without pre-training. The authors report the ability of their algorithm to detect complex structures such as sulci, the cerebellar tentorium, and the falx cerebri. Using 5-fold cross-validation on a single dataset, the authors reported dice 0.45/0.47 and Jaccard 0.3/0.31 without/with pre-training, but gave a recommendation that a pre-training is worth considering for performance improvement.**

**Rating:** 4
**Confidence:** 3

**Review:**

This is an interesting paper that considers automatic segmentation in the 3D Brain that can be of importance in real surgical situation. The paper employs standard deep learning segmentation methods for segmentation and in these techniques have been heavily used in medical imaging and it has been shown to be successful. In this sense, from machine learning aspect, it is not specifically novel.
1. There are several aspects that have not been considered, some of them mentioned by the authors. It is unclear why the authors used DeepVNet (btw, they cite the paper of DenseVNet, I guess this is the right terminology or not?). Here a natural question arises, what is the reason of specifically picking DenseVNet as there are simpler structures that can perform reasonably well (if not better, unknown?) such as V-Net (which is the segmentation equivalent for ResNet as DenseVNet is probably for DenseNets) or even simpler segmentation architectures like SegNet. The reviewer understands the space limitations, but a more thorough research should involve a performance comparison of those (DenseVNets, VNets, SegNets) and some of the simpler can be taken as benchmarks.
2. It is also unclear what particular topology of DenseVNet was picked as the depth layers size could potentially vary. Although something is specified related to the learning rate, cube size of 128^3, there are many other architectural details that are missing: depth size, kernel sizes, usage of dropout and values etc. It is worth including a tabular or visual representation of the DL architecture that is used. This also contributes to the reproducibility of the results, which could also be boosted with a potential GitHub repository on the paper.
3. Another issue that arises is whether 128^3 is enough to capture all the relevant parts of the brain.
4. The authors reported fairly low values of dice coefficients (and other correlated metrics), although suggested that visually it captures the important parts. However, from how the paper is written, they position it as 4-class (or 6-class according to the related work?) segmentation problem. They should also be more explicit in this one as well. The analysis should also involve what are the results (dice, Jaccard or other similarities) on a per-class base. In many medical segmentation problems, general dice or dice/metrics for some organs might be good, but low for other classes. Moreover, the authors should also mention the balance-ness of the class structures and how/if they deal with class imbalance problems.
5. It is not entirely clear whether the pre-training really helps as the dice and Jaccard scores are fairly similar with negligible differences. It might be beneficial for the paper to consider architecture search for better performance and to report that pre-training does not improve the performance to a big extend.

The paper is well written and related state-of-the-art is well elaborated. The reviewer suggest possible acceptance and it believes it is worth being presented at MIDL, although there are some aspects that could be improved.

**Special Issue:**

No

---

### Review · AnonReviewer3 · 2018-05-09
**Good approach to the task, but needs refining**

**Rating:** 2
**Confidence:** 2

**Review:**

This work presents a DenseVNet framework for the automatic segmentation of various structures in 3D brain ultrasound. It is testing on the publicly available RESECT dataset. As 3D ultrasound is commonly used in neurosurgery to monitor brain structures in real-time, this model may allow for reliable tracking of structures in surgery when compared to pre-operative imaging. This improves on the state of the art of registering images or manually-selected landmarks, which are prone to error, inter-observer variability, and can fail due to brain shift during surgery.

This is a good approach to providing robust registration results for real-time surgery, but more validation is required.

Pros:
- Good literature review and problem definition
- Easy to follow model
- Novel approach to the registration problem

Cons:
- Well-written, but writing quality decreases towards the end of the paper
- Not very novel from a machine-learning perspective (off the shelf model, no changes)
- Performance is measured against very coarse segmentations, yielding a Dice Coefficient of 47%. This makes it difficult to interpret this performance, as segmentations are compared to relatively poor labelling. There has to be a better way to quantify performance.
- Only looked at one model and looked at results for pre-training versus none. Why was the DenseVNet chosen? Were any other models tested?

Notes:
- In the Abstract, the model is referred to as "DeepVNet", but in-text it is "DenseVNet" -- be consistent
- Too many quantitative results in the abstract
- Figure 4 referenced before Figures 2 and 3 in-text
- In the Dataset section, it should be clarified that the T1 MRI data was only used for pre-training
- Training patches were of size 128^3 - how big are the original images? Is this a sufficient size? Was this determined experimentally?
- Should include a figure depicting the configuration of the model, or include more of this information in-text.
- Jaccard not mentioned as performance metric at beginning of Results
- Only "distance" is measured as metric at beginning of Results. Should be clarified as "Hausdorff Distance"
- "In most cases, the measures improved when we fine-tuned the network" It would be good to have a table containing the average results for scratch vs. fine-tuned. Also, it looks like pre-training yielded a very small improvement (DSC increase of 2%). Is this a significant improvement?
- Maybe data augmentation for training would improve performance by adding more examples?
- Some in-text results are shown in brackets
- In the Results: "For the refined DenseVNet, we computed ..." -- do you mean fine-tuned? Be consistent for clarity

Overall, it doesn't seem like a Dice of 47% is a good result, and comparing performance against coarse/poor segmentations seem flimsy. A stronger version of this work should potentially include results showing that the model's segmentations can be registered to MRI landmarks, as this was the prevalent issue in the introduction. Even shown qualitatively, this would lend a bit more weight to the results.

**Special Issue:**

No

---

### Review · AnonReviewer1 · 2018-05-09
**Insufficient annotated data and weak presentation.**

**Rating:** 1
**Confidence:** 2

**Review:**

This paper proposes a method for segmentation of structures in brain ultrasound, applicable particularly in intra-operative settings.  The paper is not particularly easy to read, the description of the clinical problem is convoluted and the language and sentence structure is difficult to follow at various places throughout the text.
The training data used in this case is the result of a coarse semi-automatic segmentation method which is not yet published and as such is not validated.  The images depicting the output from this method do not inspire confidence in its accuracy, and the authors admit that the results are not perfect but claim that they are "sufficient"  for training purposes.  However, my major concern with this paper is that there is no quantitative analysis comparing the final segmentations with anything other than these coarse markings.  The Dice coefficients and distance metrics presented depict relatively poor agreement.  It is impossible to tell how we should judge a method whose results have relatively poor agreement with relatively poor gold-standards.  The authors declare that they have accurate manual segmentations for "a few" images, they do not provide any quantitative analysis based on these.
In addition to the fundamental difficulties described above the paper is not well constructed and the methods are not well described.  For example the description of data simulation is unclear and it is difficult to ascertain in the end exactly which data was used for training/testing.  How much was simulated and what effect might this have?
For these reasons I am unable to recommend this paper for acceptance.

**Special Issue:**

No

---

### Decision · Program_Chairs · 2018-05-15
**Paper56 Acceptance Decision**

Reject